# Association of [^68^Ga]Ga-PSMA-11 PET/CT Metrics with PSA Persistence Following Radical Prostatectomy in Patients with Intermediate- and High-Risk Prostate Cancer

**DOI:** 10.3390/diagnostics15030301

**Published:** 2025-01-27

**Authors:** Juan J. Rosales, Vicky Betech-Antar, Fernando Mínguez, Edgar F. Guillén, Elena Prieto, Gemma Quincoces, Carmen Beorlegui, María Dolores Fenor de la Maza, Fernando Díez-Caballero, Bernardino Miñana, José Luis Pérez-Gracia, Macarena Rodríguez-Fraile

**Affiliations:** 1Department of Nuclear Medicine, Clínica Universidad de Navarra, 31008 Pamplona, Spain; vbetechanta@unav.es (V.B.-A.); fminguez@unav.es (F.M.); gquinfer@unav.es (G.Q.); mrodriguez@unav.es (M.R.-F.); 2Department of Nuclear Medicine, Clínica Universidad de Navarra, 28027 Madrid, Spain; eguillenv@unav.es; 3Department of Medical Physics, Clínica Universidad de Navarra, 31008 Pamplona, Spain; eprietoaz@unav.es; 4Service of Planning, Evaluation and Knowledge Management, Department of Health, Government of Navarra, 31008 Pamplona, Spain; mcbeor@gmail.com; 5Department of Medical Oncology, Clínica Universidad de Navarra, 28027 Madrid, Spain; mfenordelam@unav.es; 6Department of Urology, Clínica Universidad de Navarra, 31008 Pamplona, Spain; fdcaballero@unav.es; 7Department of Urology, Clínica Universidad de Navarra, 28027 Madrid, Spain; bminana@unav.es; 8Department of Medical Oncology, Clínica Universidad de Navarra, 31008 Pamplona, Spain; jlgracia@unav.es

**Keywords:** PSMA-PET, prostate cancer, initial staging, PSA persistence, volume-based metrics

## Abstract

**Background/Objectives:** The aim of this study was to determine whether semiquantitative and volume-based metrics obtained from [^68^Ga]Ga-PSMA-11 PET/CT (PSMA-PET) scans before radical prostatectomy (RP) are associated with PSA persistence after surgery in patients with intermediate- (IR) and high-risk (HR) prostate cancer (PCa). **Methods:** We included 118 consecutive patients (IR = 57; HR = 61) with PCa with a PSMA-PET for initial staging and underwent subsequent RP. Clinical parameters and PSMA-PET metrics in the prostate were obtained to determine the following measurements: SUVmax, SUVmean, Target-to-Background Ratios (TBRs), Prostate Molecular Tumor (pMTV), Prostate Total Lesion Activity (pTLA), Prostate Volume (pV), and Prostate Disease Burden (pDB). The association of PSMA-PET metrics parameters before RP and PSA persistence were analyzed by multivariate logistic regression. **Results:** SUVmax and volume-based PSMA-PET metrics were significantly higher in patients with ISUP Grade 3–5 vs. ISUP Grade 1–2, and only pMTV, pTLA, and pDB were found to be significantly higher in HR patients, as compared with the IR group. During follow-up, 23 patients showed PSA persistence. pMTV, pTLA, and pDB were significantly higher among patients presenting PSA persistence after RP than in patients with undetectable PSA. Multivariate logistic regression analysis found that lymph node infiltration and pTLA were independent predictors for PSA persistence. A cut-off point of ≥25.1 allowed the best discrimination for PSA persistence (OR: 7.4; IQR: 1.4–39.1; *p* < 0.05). **Conclusions:** The identified association between PSA persistence and prostate TLA of PSMA-PET at initial staging highlights its potential as a valuable tool to improve risk prediction in prostate cancer patients. Further research is needed to confirm these results.

## 1. Introduction

Prostate cancer (PCa) is the most common cancer-related death in men worldwide, and the second leading cause of cancer death in men [1]. Even with early intervention in localized disease, 20–50% of patients will experience biochemical recurrence (BCR) within the first decade after radical prostatectomy (RP) or external beam radiotherapy [2,3,4,5]. The use of less sensitive and quite unspecific diagnostic techniques including computed tomography (CT) and bone scintigraphy (BS) compared to current molecular imaging modalities, to detect local and distant disease at initial staging may partially influence the rates of BCR following radical treatment [6,7]. Positron emission tomography targeting prostate-specific membrane antigen (PSMA-PET) has emerged in recent years as the reference standard examination not only for BCR recurrence but also for initial staging in patients with intermediate- and high-risk PCa according to the latest guidelines from the European Association of Urology (EAU) with better accuracy and fewer equivocal results as compared with conventional imaging modalities (CT and BS) [8,9,10,11,12].

Currently, initial risk and BCR assessment is still based exclusively on clinical and histopathological parameters (e.g., age at diagnosis, preoperative PSA levels, Gleason Score, ISUP grade, or clinical T stage) [13,14]. Therefore, current risk models might require a more holistic approach that enables more accurate and reliable predictions. Recently, nomograms using PSMA-PET in combination with Prostate Cancer Molecular Imaging Standardized Evaluation (PROMISE) criteria have demonstrated accurate risk stratification for high- and low-risk groups for overall survival in early and late stages of PCa, with equal or superior prediction accuracy than the models proposed by either EAU, the National Comprehensive Cancer Network (NCCN), or the International Staging Collaboration for Cancer of the Prostate (STARCAP) [15].

The intensity of intraprostatic PSMA uptake, quantified by the Maximum Standardized Uptake Value (SUVmax), has been identified as a significant predictor of BCR [16,17,18,19,20]. Additionally, whole-body PSMA-PET parameters such as Molecular Tumor Volume (MTV), SUVmean, and Total Lesion Activity (TLA) have been investigated for their roles in patient selection and response assessment to radioligand therapy with ^177^Lu[Lu]-PSMA-617 [21,22]. Despite these advancements, the actual evidence on the prognostic value of preoperative volumetric PSMA-PET parameters in predicting BCR, specifically in the PSA persistence setting is still lacking.

Therefore, the aim of this study was to determine whether the volumetric parameters of the prostate gland obtained from PSMA-PET scans before RP are associated with PSA persistence in patients with intermediate- and high-risk PCa.

## 2. Materials and Methods

### 2.1. Study Design and Participants

Consecutive patients with treatment naïve PCa staged with [^68^Ga]Ga-PSMA-11 PET/CT studied at our center from December 2018 until December 2023 were analyzed in this investigation. The inclusion criteria included (a) biopsy-proven PCa, (b) intermediate- or high-risk PCa as defined by EAU classification [12], (c) positive PSMA expression in the primary tumor according to PROMISE V1 criteria [23], and (d) subsequent treatment with RP with or without pelvic lymph node dissection (PLND). Patients who had (a) history of hormonal therapy before surgery; (b) extrapelvic lymph nodes, bone, and visceral metastasis on PSMA-PET; or (c) low-grade PSMA expression in primary tumor (Score 0–1) were excluded from the analysis. Figure 1 shows the study design.

The protocol was approved by our Institution’s ethics committee (protocol number: 2021.201), and the entire investigation was carried out in accordance with the ethical standards laid down in the 1964 Declaration of Helsinki and all its subsequent revisions. Informed consent was obtained from all patients.

### 2.2. [^68^Ga]Ga-PSMA-11 PET/CT Protocol

PSMA-PET was performed according to the EANM/SNMMI procedure guidelines [24]. The images were acquired in two tomographs (Siemens Biograph mCT 64 and Siemens Vision 600, Siemens, Knoxville, TN, USA). Patients received an intravenous injection of 155 ± 45 MBq of [^68^Ga]Ga-PSMA-11. Vertex to mid-thigh PET/CT scans began 69.6 ± 15.1 min after injection, with 3 min per bed position for PET imaging and a low dose non-enhanced CT (Care Dose 4D with a quality reference of 80 mAs at 120 kV). PET data were reconstructed using an iterative algorithm with 3 iterations, 21 subsets (OSEM 3–21) into a 200 × 200 matrix. For each acquisition, two different reconstructions were performed: a high-quality reconstruction, optimized for lesion detection (PSF + TOF, 2 mm Gaussian filter), and a reconstruction in compliance with EARL Standard 1 for quantification (TOF, 5 mm Gaussian filter).

### 2.3. Image Analysis

PSMA-PET images were reviewed by two nuclear medicine physicians (MRF and JJR). Tumor and nodal staging on PSMA-PET were reported based on the criteria featured in the E-PSMA guidelines [25]. The quantitative analysis of the molecular parameters from the PSMA-PET imaging was conducted by the second nuclear physician (JJR). A spherical volume of interest (VOI) was drawn in all the pathological PSMA uptake in the prostate gland to encompass the entire target lesion and isoactivity contours were automatically generated. A threshold percentage cut-off range between 40 and 45% was used to provide the best visual contouring of the boundaries of the tumor in the prostate gland, as previously described by other authors [26,27]. Prostate Volume (pV) was obtained through the automated segmentation of the prostate gland on CT images using work-in-progress syngo.via MI General Anatomy Segmentation (Siemens Healthineers, Knoxville, TN, USA) (Figure 2).

PSMA-PET metrics in the prostate gland were calculated and named as follows: tumor SUVmax, tumor SUVmean, Prostate Molecular Tumor Volume (pMTV), Prostate Volume (pV), Prostate Total Lesion Activity (pTLA), and Prostate Disease Burden (pDB), which was calculated using the formula (pMTV × 100/pV). In patients with miT3b, the infiltration of the seminal vesicles was included in the prostrate volume-based metrics.

Although the specificity of PSMA-PET is high in nodal staging [8], PSMA-positive lymph nodes on PSMA-PET were not included in prostate volume-based metrics for two reasons. Firstly, all patients included in our series were surgically treated upfront, and a low incidence of nodal tumor involvement was expected. Secondly, this exclusion aimed to prevent the potential influence of false positive lymph nodes on volume-based metrics, which could lead to an overestimation of MTV.

Tumor-to-normal organ ratios (TBRs) were also calculated through the division of the tumor SUVmax by the SUVmean of the liver, spleen, mediastinal blood pool, and salivary glands, using a 1cc spherical VOI in each region.

### 2.4. Outcome

PSA persistence was defined as failure of PSA to fall to undetectable levels (<0.1 ng/mL) in 4–8 weeks following RP, according to the latest version of the NCCN and EAU guidelines [12,28].

### 2.5. Statistical Analysis

Qualitative variables were described by frequency and percentage. All continuous variables were tested for normality using the Kolmogorov–Smirnov test. Data are presented as median values and interquartile range (IQR). Patients were compared using the nonparametric Mann–Whitney U test. We performed univariate and multivariate logistic regression analyses to identify prostate PSMA-PET metrics associated with PSA persistence. The results of logistic regression were expressed as odds ratios (OR) and 95% confidence intervals (CI). A receiver operating characteristic (ROC) curve was computed to determine optimal cut-off values for PSMA-PET metrics to predict the likelihood of PSA persistence, with the total area under the curve (AUC), its 95% CI, and the Se and Sp at the optimal cut-off according to the Youden index (maximum of [Se + Sp − 1]) calculated as indicators of overall goodness of the model. Values of *p* < 0.05 were considered statistically significant. All statistical analyses were performed using IBM SPSS Statistics for Windows, version 29 (IBM Corp., Armonk, NY, USA).

## 3. Results

### 3.1. Patient Characteristics

We included 118 consecutive patients with biopsy-proven PCa who underwent RP following PSMA-PET for initial staging. Fifty-seven patients (48.3%) were classified as intermediate-risk, and sixty-one (51.7%) as high-risk, according to the EAU risk classification [12]. The clinicopathological characteristics of patients are summarized in Table 1.

### 3.2. Prostate PSMA-PET Metrics by Baseline Risk Classification (Intermediate- vs. High-Risk)

From all the PSMA-PET metrics obtained, only volumetric parameters (pMTV, pTLA, and pDB) were found to be significantly increased among patients in the high-risk group, as compared with the intermediate-risk group (*p* < 0.05). The high-risk group exhibited a median pMTV of 4.6 cc, a median pTLA of 29.4, and a median pDB of 9.2%. In comparison, the intermediate-risk group showed median values of 3.6 cc for pMTV, 19.1 for pTLA, and 6.8% for pDB, respectively. No significant differences were observed in the rest of the PSMA-PET metrics between both groups (Table 2).

### 3.3. Prostate PSMA-PET Metrics by Final ISUP Grade (ISUP 1–2 vs. ISUP 3–5)

The median tumor SUVmax values were significantly higher in patients presenting with ISUP Grade 3–5, as compared with the ISUP Grade 1–2 group (11.6 vs. 8.8, respectively, *p* = 0.03). These significant differences were also observed for volume-based PSMA-PET metrics as shown in Figure 3. No significant differences were observed in terms of tumor SUVmean, TBRs or pV between both groups.

### 3.4. Prostate PSMA-PET Metrics by Seminal Vesicles Infiltration

Histopathological analysis showed seminal vesicle infiltration in 20 of 118 patients (16.9%). In this group, median values for pMTV, pTLA, and pDB were significantly higher: 7.7 cc (IQR: 5.1–14.4), 39.3 (IQR: 27.9–56.6), and 12.1% (IQR: 8.8–22.4), respectively. In contrast, patients without seminal vesicle infiltration exhibited median values of 3.6 cc (IQR: 2.1–5.6), 19.3 (IQR: 11.8–30.6), and 6.8% (IQR: 3.8–11.1) for pMTV, pTLA, and pDB, respectively. PSMA-PET detected infiltration of seminal vesicles in 15 of the 20 patients (75%) in which infiltration was confirmed by histopathology.

### 3.5. Prostate PSMA-PET Metrics by Node Infiltration Status

PLND was performed in 87 of 118 patients, with 68 undergoing standard PLND and 19 undergoing extended PLND. Among these, 16 patients were found to have pelvic nodal infiltration based on histopathological analysis (pN1), and in 12 of them (75%), node infiltration was confirmed by positive PSMA expression (true positive miN1). Of the 31 patients who did not undergo PLND, none exhibited persistent PSA levels after surgery; therefore, they were classified as true N0 for comparison. Patients with pN1 and true positive miN1 disease had significantly higher values of pMTV, pTLA, and pDB, as compared not only with the N0 group (pN0 and true negative miN0) but also with the group of patients presenting false positive miN1 disease (Table A1). No significant differences were observed in the remaining PSMA-PET metrics.

### 3.6. Prostate PSMA-PET Metrics by PSA Status Following RP (Undetectable vs. Persistent)

During follow-up, 23 patients (19.5%) presented persistent PSA levels, while 95 patients (80.5%) exhibited undetectable PSA levels following RP. Among patients presenting persistent PSA, 19 were classified as high risk, and 4 as intermediate risk.

Interestingly, volumetric PSMA-PET parameters in the prostate gland, such as pMTV, pTLA, and pDB, were significantly higher among patients presenting PSA persistence. For pMTV, the analysis showed a median of 8.6 cc in patients with PSA persistence, whereas in subjects presenting undetectable levels of PSA following surgery, the median was 3.6 cc (*p* < 0.001). The median values of pTLA and pDB were also significantly higher in patients presenting PSA persistence following surgery (44.2 and 13.2%, respectively) than in patients presenting negative PSA (19.3 and 6.9%, respectively; *p* < 0.05 each). Although tumor SUVmax and tumor SUVmean values were also higher in patients presenting persistent PSA levels, these differences did not reach statistical significance. Additionally, no significant differences were noted in TBRs or pV between both groups (Table 3).

### 3.7. ROC Analysis and Logistic Regression of PSMA-PET Metrics as Predictors of PSA Persistence

The usefulness of employing the PSMA-PET metrics (pMTV, pTLA, and pDB) before surgery to discriminate patients who had a PSA persistence after RP was established by means of ROC curves. The corresponding ROC curve plots are displayed in Figure 4. The significant cut-off points for the curves had the following values: 4.7 cc for pMTV, 25.1 for pTLA, and 9.1% for pDB. Table 4 shows the rest of the diagnostic indices from ROC analysis for each parameter.

The univariate logistic regression analysis showed that pN1, pMTV, pTLA, and pDB correlated with the presence of persistent PSA levels following surgery, while the multivariate logistic regression analysis found that only pTLA (OR: 7.4; IQR: 1.4–39.1; *p* = 0.017) and pN1 (OR: 15.6; IQR: 3.3–73.6; *p* = 0.001) were independent predictors for PSA persistence. The results of the univariate and multivariate analysis are summarized in Table 5.

## 4. Discussion

The aim of this study was to investigate classical and volumetric parameters of PSMA-PET in the prostate gland in patients with intermediate- and high-risk PC at initial staging before RP, and their possible association with PSA persistence.

Several global risk classification systems are being used for stratifying newly diagnosed PCa patients, including the EAU, NCCN, Cambridge Prognostic Group, and CAPRA score [12,28,29,30,31]. These systems integrate prognostic factors like PSA levels, clinical tumor stage, ISUP grade groups, and Gleason Scores. However, discrepancies between biopsy and final Gleason Scores have been documented, potentially leading to incorrect initial risk group classification [32,33]. Consequently, additional clinical, genetic, and molecular imaging biomarkers, including radiomics, are essential not only to improve PCa stratification but also for assessing the risk of BCR. Recently, the combination of PSMA-PET and PROMISE criteria has demonstrated accurate stratification among risk groups for overall survival in early- and late-stage PCa when compared with established clinical nomograms in a large PCa dataset [15].

At initial staging, our investigation revealed that volume-based PSMA-PET metrics in the prostate gland (pMTV, pTLA, and pDB) were significantly higher, not only among high-risk patients as compared with intermediate-risk patients but also in patients with pN1 disease, when compared with those with negative lymph nodes by either histopathology or PSMA-PET. This suggests that pMTV, pTLA, and pDB may help to discriminate patients by risk group and lymph node status.

One interesting finding is that we observed the same tendency in patients with false positive miN1 disease, which presented lower pMTV, pTLA, and pDB than patients presenting true positive N1 disease.

Recently, Incesu et al. reported that PLDN does not impact short-term oncologic outcomes in patients with intermediate-risk PCa and negative nodal status on PSMA PET, suggesting that it may be avoided in this clinical setting. While higher values of pMTV, pTLA, and pDB are associated with nodal involvement in histopathology and could aid in guiding surgical treatment decisions (e.g., extending the template of dissection), the utility of PSMA PET volume-based parameters in guiding surgical decisions still needs to be validated [34].

Numerous studies have shown that there is a notable increase in the expression of PSMA in patients with ISUP grade group ≥ 2 [35,36]. Since SUVmax measurements often represent values restricted just to one pixel, and such values could be influenced by the effects of noise and image resolution, its ability to assess the molecular characteristics of the entire tumor tissue is limited [37,38]. However, previous studies have highlighted the usefulness of SUVmax in different clinical PCa settings. Consistent with the results reported by Rogic et al. and Bodar et al. [39,40], our analysis revealed that patients diagnosed with ISUP grade group 1–2 exhibited significantly lower intraprostatic SUVmax values when compared to those ISUP grade group 3–5 (Figure 3). This finding underscores a potential association between SUVmax and tumor aggressiveness.

The use of volume-based PSMA metrics has also been studied in patients presenting BCR and metastatic PCa. Studies have shown that tumor burden, as quantified through these volumetric approaches, exhibits a statistically significant correlation with not only serum PSA levels but also with clinical outcomes such as treatment response, treatment failure, and BCR [41,42]. To the best of our knowledge, our study is the first one to assess the association between volume-based PSMA-PET metrics in the prostate gland and PSA persistence. Uhlman et al. previously evaluated this same concept by studying prostate specimens after surgery, reporting that tumor volume, total prostatic infiltration, and prostate volume were predictors of BCR [43]. We achieved similar results, observing that pMTV, pTLA, and pDB were significantly higher in patients with PSA persistence than in those who remained with undetectable PSA levels following RP. In this respect, the comprehensive assessment of the tumor’s clinical, pathological, and molecular parameters through volumetric methods obtained by PSMA-PET might provide additional crucial information for making therapeutic decisions. Specifically, a pTLA ≥ 25.1 on PSMA-PET at initial staging was an independent factor for PSA persistence in multivariate analysis, suggesting that high tumor burden in the prostate gland before surgery could serve as an additional risk factor. Multicenter validation studies are needed to determine whether these parameters can be integrated into existing validated nomograms.

Our results provide insights into the relevance of evaluating PSMA-PET volume-based metrics. It could be hypothesized that high tumor burden in the prostate gland assessed by PSMA-PET may be associated with early biochemical failure in localized PCa following RP and that this imaging parameter could potentially be used for risk stratification.

While our study provides valuable insights, it is subject to certain limitations, including the sample size (*n* = 118), the single-center setting, and its retrospective design. Another limitation, which could also be considered a strength, is the exclusion of nodal tumor burden from the calculated tumor volumes. This approach was intentional, as we aimed to prevent the possible false positive miN1 cases from interfering with the total prostate volume measurements. By focusing solely on disease within the prostate gland, we believe this methodology offers a straightforward and reproducible approach using current segmentation tools. Nonetheless, future research should aim to validate and expand upon these findings through larger cohorts and prospective study designs.

## 5. Conclusions

In conclusion, our study adds evidence to the evolving landscape of PCa management by exploring the value of volume-based PSMA-PET metrics. The identified association between PSA persistence and prostate TLA of PSMA-PET at initial staging highlights its potential as a valuable tool to improve the accuracy and reliability of risk prediction models in PCa patients. Further research is warranted to fully integrate these volumetric parameters into clinical practice.

## Figures and Tables

**Figure 1 diagnostics-15-00301-f001:**
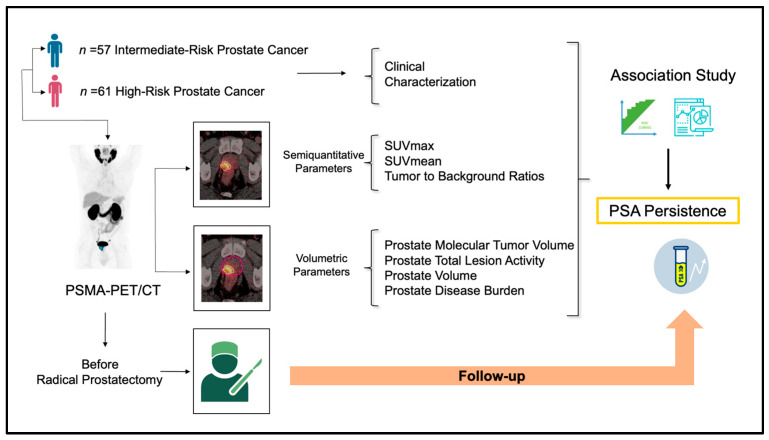
Study design.

**Figure 2 diagnostics-15-00301-f002:**
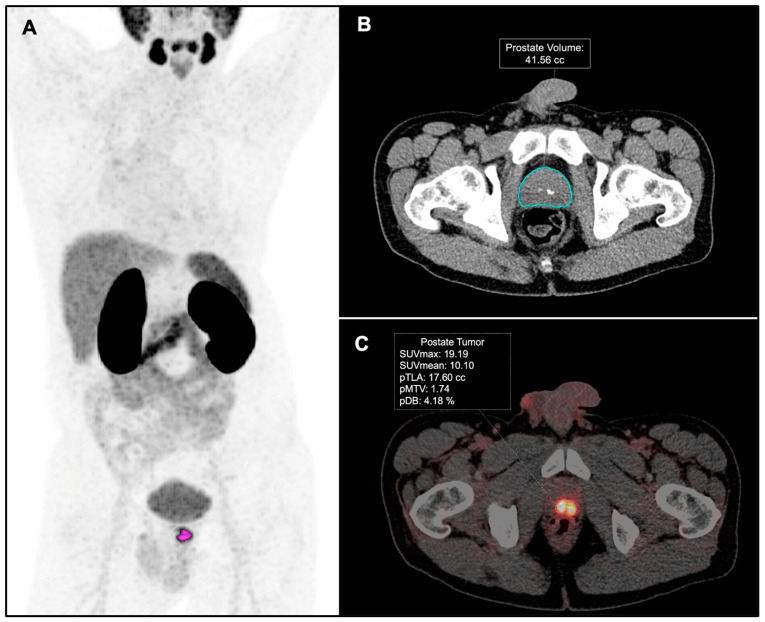
Exemplary patient with high-risk PCa. (**A**) PET-PSMA maximum intensity projection. (**B**) Automatic segmentation of the prostate gland on axial CT images. (**C**) PSMA-PET metrics in primary tumor.

**Figure 3 diagnostics-15-00301-f003:**
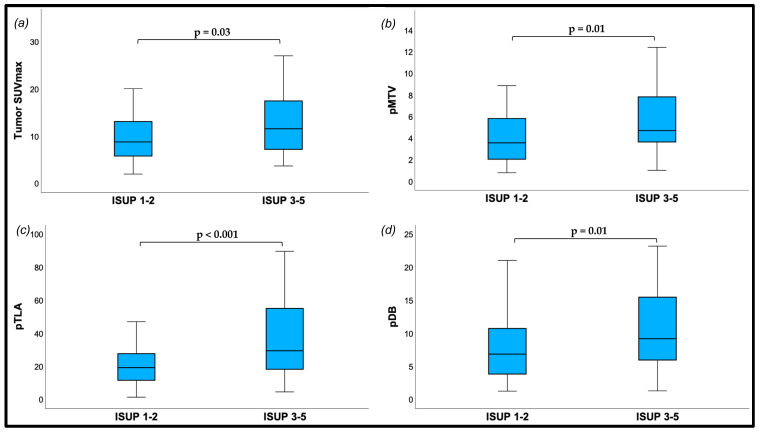
Boxplots displaying differences in PSMA-PET metrics in patients with ISUP Grade 1–2 and patients with ISUP Grade 3–5. (**a**) Tumor SUVmax (median: 8.8 vs. 11.6, *p* = 0.03); (**b**) pMTV (median: 3.6 vs. 4.7, *p* = 0.01); (**c**) pTLA (median: 19.2 vs. 29.4, *p* < 0.001); (**d**) pDB (median: 6.8 vs. 9.2, *p* = 0.01).

**Figure 4 diagnostics-15-00301-f004:**
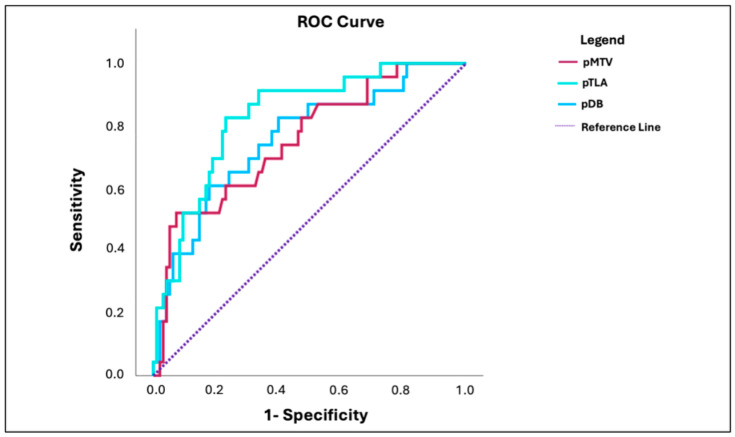
ROC curves demonstrating the usefulness of PSMA-PET metrics for predicting the presence of PSA persistence.

**Table 1 diagnostics-15-00301-t001:** Clinicopathological characteristics of patients.

Characteristics	*n* (%)—Median (IQR)
EAU Risk Classification	
Intermediate-Risk	57 (48.3)
High-Risk	61 (51.7)
Age (years)	65 years (61–70)
Biopsy ISUP Grade Group (Gleason Score)	
Group 1 (3 + 3)	24 (20.3)
Group 2 (3 + 4)	28 (23.7)
Group 3 (4 + 3)	21 (17.8)
Group 4 (8)	29 (24.6)
Group 5 (9–10)	16 (13.6)
PSA at PET (ng/mL)	10.8 ng/mL (7.1–16.9)
Time between biopsy and PSMA-PET	34 days (23–63)

**Table 2 diagnostics-15-00301-t002:** Baseline PSMA-PET metrics by EAU risk classification (intermediate- vs. high-risk).

PSMA-PET Metrics	Intermediate Risk Median (IQR)	High Risk Median (IQR)	*p* *
SUVmax	9.4 (5.8–12.8)	10.1 (6.3–18.0)	0.195
SUVmean	5.9 (3.6–7.6)	5.8 (3.7–9.0)	0.503
Liver SUVmean	4.6 (3.7–5.4)	4.6 (3.8–5.7)	0.650
TBR-liver	1.5 (1.0–2.1)	1.6 (1.0–2.8)	0.482
TBR-spleen	1.3 (0.7–2.0)	1.1 (0.7–2.0)	0.959
TBR-blood pool	6.7 (4.3–9.2)	6.7 (4.5–11.1)	0.275
TBR-salivary glands	0.5 (0.3–0.7)	0.6 (0.3–1.1)	0.320
pMTV (cc)	3.6 (1.9–6.0)	4.6 (3.0–6.6)	**0.022**
pTLA	19.1 (11.8–25.4)	29.4 (15.3–54.4)	**0.001**
pDB (%)	6.8 (3.4–9.6)	9.2 (5.6–14.5)	**0.007**
pV (cc)	52.9 (41.2–70.5)	51.3 (40.1–6.9)	0.504

* Values in bold show a *p* < 0.05.

**Table 3 diagnostics-15-00301-t003:** Baseline PSMA-PET metrics: undetectable PSA after RP vs. PSA persistence.

PSMA-PET Metrics	Undetectable PSA After RP Median (IQR)	PSA Persistence Median (IQR)	*p*-Value *
Tumor SUVmax	9.4 (5.8–13.1)	11.6 (8.7–22.8)	0.071
Tumor SUVmean	5.2 (3.7–7.6)	6.33 (5.4–12.3)	0.062
Liver SUVmean	4.6 (3.8–5.6)	4.4 (3.5–5.1)	0.211
TBR-liver	1.5 (1.0–2.1)	2.2 (1.2–2.8)	0.083
TBR-spleen	1.1 (0.7–2.0)	1.58 (1.2–2.8)	0.065
TBR-blood pool	6.4 (4.5–9.2)	9.7 (5.2–11.3)	0.090
TBR-salivary glands	0.5 (0.3–0.8)	0.71 (0.4–1.2)	0.050
pMTV	3.6 (2.1–5.8)	8.6 (4.1–14.1)	**<0.001**
pTLA	19.3 (11.8–27.7)	44.2 (30.8–74.5)	**<0.001**
pDB	6.9 (3.9–10.1)	13.2 (8.7–22.3)	**0.001**
pV	51.4 (40.5–69.0)	54.00 (43.8–72.8)	0.750

* Values in bold show a *p* < 0.05.

**Table 4 diagnostics-15-00301-t004:** Diagnostic indices from ROC analysis of PSMA-PET metrics for PSA persistence.

PET/CT Metrics	Cut-Off	Sensitivity % (95% CI)	Specificity % (95% CI)	YI	AUC	*p*-Value *
pMTV	4.7	69.6 (47.1–86.8)	64.2 (53.7–73.8)	0.338	0.754 (0.643–0.865)	**<0.001**
pTLA	25.1	91.3 (71.9–98.9)	66.3 (55.9–75.7)	0.576	0.831 (0.743–0.918)	**<0.001**
pDB	9.1	73.9 (51.6–89.7)	66.3 (55.9–75.7)	0.402	0.758 (0.646–0.869)	**<0.001**

* Values in bold show a *p* < 0.05.

**Table 5 diagnostics-15-00301-t005:** Univariate and multivariate logistic regression results for predicting PSA persistence (*n* = 23).

Parameters (*n*)	Univariate Analysis		Multivariate Analysis	
	OR (95% CI)	*p*-Value	OR (95% CI)	*p*-Value *
pN1 (16)	39.8 (9.6–164.1)	<0.001	15.6 (3.3–73.6)	**0.001**
pMTV > 4.7 (50)	4.1 (1.5–10.9)	0.005	0.8 (0.21–3.5)	0.842
pTLA > 25.1 (53)	20.6 (4.5–93.7)	<0.001	7.4 (1.4–39.1)	**0.017**
pDB > 9.1 (49)	5.5 (2–15.2)	<0.001	2.64 (0.58–12)	0.209

* Values in bold show a *p* < 0.05.

## Data Availability

The data presented in this study are available on request from the corresponding author. The data are not publicly available due to privacy.

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
