# Peer review of "Association of [68Ga]Ga-PSMA-11 PET/CT Metrics with PSA Persistence Following Radical Prostatectomy in Patients with Intermediate- and High-Risk Prostate Cancer"

_diagnostics, 2025, doi:10.3390/diagnostics15030301_

Round 1
Reviewer 1 Report
Comments and Suggestions for Authors
The authors present a very interesting study on the value of PSMA-PET as a predictive imaging technique prior to radical prostatectomy.
I would see two issues discussed more in detail:
- what could be the advantage for the surgeon having such an imaging prior to surgery?
Does it help to find suspicious lymph nodes (ie. to extent the template of dissection)?
Could it be helpful to change the surgical approach (ie. no nerve-sparing in case of advanced disease)?
Otherwise, the conclusions of the study would be clear, but do not provide anything further to improve the results of surgery?
Would it make also sense to use the imaging results as an indicator for Neo-adjuvant chemo-/hormonal therapy?
Reviewer 2 Report
Comments and Suggestions for Authors
Dear Editors,
The manuscript entitled: "Association of 68Ga[Ga]-PSMA-11 PET/CT metrics with PSA persistence following radical prostatectomy in patients with intermediate- and high-risk prostate cancer." is a well-written original article investigating the potential role of preoperative prostate cancer primary tumor PSMA PET parameters in predicting the PSA persistance following radical prostatectomy.
I have a few major complaints to the text and multiple minor issues listed below, which should be addressed by the authors, before the final acceptance and publication. I would kindly recommend the editors to revise the corrected version of the manuscript and reconsider the acceptance.
Line 48: cancer-related death
51 less sensitive than what?
52 - please use lower case letters for computed tomogaphy, bone scan, etc.
Fig. 1. Characterization
104 vertex to mid-thigh
114 nuclear medicine physicians
132 false positive lymph nodes at PSMA PET - they are very rare, please comment on that and mention that the specificity of PSMA PET in detecting LN mets is very high.
131-133 - the paragraph is not fully clear, please specify what do you mean by 'negative influence'.
Reference 12 should by cited properly.
174 (...) and 6.8 % for pDB, respectively.
Table 2 - put the significant p values in bold and explain that in the table subtitles. In other tables - as well.
181 (...) (11.6 vs 8.8, respectively, p = 0.03).
Please, remember to use the word 'respectively', whenever it is necessary.
Fig. 3 - too small fonts, difficult to read.
Table 5 - please, show also the non-statistically significant values (with their p values) for comparison.
261 - for improve -> to improve
Round 2
Reviewer 2 Report
Comments and Suggestions for Authors
Dear Editors,
I accept most of the authors' answers to my questions and concerns. The authors have done a good job!
I have only one remaining concern about the Fig. 3 - there are still some small numbers, which I recommend to either enlarge or delete, to make the diagrams clearer. Also, I recommend to paste the p values directly into the Figure 3 graphs.
Otherwise, I accept the manuscript in the present form.
Author Response
Dear Editors,
I accept most of the authors' answers to my questions and concerns. The authors have done a good job!
I have only one remaining concern about the Fig. 3 - there are still some small numbers, which I recommend to either enlarge or delete, to make the diagrams clearer. Also, I recommend to paste the p values directly into the Figure 3 graphs.
Otherwise, I accept the manuscript in the present form.
Response:
Dear Reviewer, thank you very much for your valuable comments. Based on your suggestions, we have removed the small numbers in Fig. 3 to improve clarity and have also added the p-values.
